# Structural color in *Junonia* butterflies evolves by tuning scale lamina thickness

Rachel C Thayer[1]*, Frances I Allen[2,3], Nipam H Patel[1,4]

[1]Department of Integrative Biology, University of California, Berkeley, Berkeley, United States; [2]Department of Materials Science and Engineering, University of California, Berkeley, Berkeley, United States; [3]California Institute for Quantitative Biosciences, University of California, Berkeley, Berkeley, United States; [4]Marine Biological Laboratory, Woods Hole, United States

**Abstract** In diverse organisms, nanostructures that coherently scatter light create structural color, but how such structures are built remains mysterious. We investigate the evolution and genetic regulation of butterfly scale laminae, which are simple photonic nanostructures. In a lineage of buckeye butterflies artificially selected for blue wing color, we found that thickened laminae caused a color shift from brown to blue. Deletion of the *optix* patterning gene also altered color via lamina thickening, revealing shared regulation of pigments and lamina thickness. Finally, we show how lamina thickness variation contributes to the color diversity that distinguishes sexes and species throughout the genus *Junonia*. Thus, quantitatively tuning one dimension of scale architecture facilitates both the microevolution and macroevolution of a broad spectrum of hues. Because the lamina is an intrinsic component of typical butterfly scales, our findings suggest that tuning lamina thickness is an available mechanism to create structural color across the Lepidoptera.

## Introduction

Structural colors are both visually delightful and abundant in nature. Organisms deploy structural colors to display hues for which they lack pigments (frequently blues and greens), to create specific optical effects such as iridescence or light polarization, and to mediate ecological interactions, including intraspecific signaling and camouflage. Unlike pigmentary color, which is caused by molecules that selectively absorb certain wavelengths of light, structural colors result from the constructive and destructive interference of light as it interacts with nanoscale, precisely-shaped physical structures that are made of a high refractive index material (e.g. keratin, chitin, or cellulose).

Despite the clear importance of structural color for living systems, the biological production of structural colors has long eluded characterization (*Cuthill et al., 2017*). Many experimental techniques depend on harnessing variation to dissect biological processes, but photonic structures are so small that quantitatively measuring variation in their dimensions is technically demanding, especially for high-throughput sampling, detecting subtle variation that may segregate within populations, or analyzing over developmental time in vivo. The color itself is easier to quantify, but has limited utility as a proxy for nanostructural dimensions, since structural colors and pigments often co-occur and covary. While recent studies (*Parnell et al., 2018*; *Matsuoka and Monteiro, 2018*; *Brien et al., 2018*) have made early headway toward describing genetic regulation of structural colors, much work remains to decipher the evolutionary, developmental, and genetic bases of structural coloration, and lab-tractable systems with intraspecific variation in structural coloration are needed. We present a promising system, the butterfly genus *Junonia,* with extensive variation in a simple structural color, and show how structural simplicity is a tactical advantage when seeking to unravel mechanisms for the biological production of nanostructures.

*For correspondence:
thayerr@berkeley.edu

Competing interests: The authors declare that no competing interests exist.

**eLife digest** From iridescent blues to vibrant purples, many butterflies display dazzling 'structural colors' created not by pigments but by microscopic structures that interfere with light. For instance, the scales that coat their wings can contain thin films of chitin, the substance that normally makes the external skeleton of insects. In slim layers, however, chitin can also scatter light to produce color, the way that oil can create iridescence at the surface of water.

The thickness of the film, which is encoded by the genes of the butterfly, determines what color will be produced. Yet, little is known about how common thin films are in butterflies, exactly how genetic information codes for them, and how their thickness and the colors they produce can evolve.

To investigate, Thayer et al. used a technique called Helium Ion Microscopy and examined the wings of ten related species of butterflies, showing that thin film structures were present across this sample. However, the different species have evolved many different structural colors over the past millions of years by changing the thickness of the films.

Next, Thayer et al. showed that this evolution could be reproduced at a faster pace in the laboratory using common buckeye butterflies. These insects mostly have brown wings, but they can have specks of blue created by thin film structures. Individuals with more blue on their wings were mated and over the course of a year, the thickness of the film structures increased by 74%, leading to shiny blue butterflies. Deleting a gene called *optix* from the insects also led to blue wings. *Optix* was already known to control the patterns of pigments in butterflies, but it now appears that it controls structural colors as well.

From solar panels to new fabrics, microscopic structures that can scatter light are useful in a variety of industries. Understanding how these elements exist and evolve in organisms may help to better design them for human purposes.

In butterflies, photonic nanostructures occur within the architecture of scales. Scales are the fundamental coloration unit on butterfly wings and have a Bauplan consisting of a grid of ridges and crossribs, supported by a lower lamina that is a simple plane (*Figure 1A*). Scales are composed of chitin and may also have embedded pigments. Intricate architecture and a high refractive index make scales a pliable substrate for photonic innovations, and indeed scales have been evolutionarily elaborated in many ways for impressive optical effects (*Ghiradella, 1985*). Even the simplest butterfly scales can produce structural color, via the lower lamina acting as a thin film reflector. Thin films are the simplest photonic structure and consist of a layer of high refractive index material, on the order of hundreds of nanometers thick, surrounded by a material with a contrasting refractive index, such as air (*Figure 1B*). Light is reflected from each surface of the film, and these two reflections interfere with each other. If the two reflections remain in phase, which depends on the extra distance traveled through the film and the wavelength, then they interfere constructively to produce observable color (*Mason, 1927*; *Yeh et al., 1978*). Conversely, wavelengths (colors) that undergo destructive interference have decreased brightness.

While it is known that the thickness of the lower lamina is one parameter that controls structural color wavelength (*Stavenga et al., 2014*) and that thickness can respond to artificial selection in the laboratory (*Wasik et al., 2014*), it is not known how general this mechanism is in natural evolution. It is also unknown how lamina structural colors are genetically regulated and whether any recognized butterfly wing patterning genes regulate lamina thickness. Here, we use mutants with deletion of the *optix* wing patterning gene, artificial selection on wing color, and genus-wide wing color variation to test the role of lamina thickness in generating butterfly color. We show that butterflies in the genus *Junonia* thoroughly exploit the relationship between film thickness and color, using the thin films necessarily present in their scales to produce a broad spectrum of hues by tuning lamina thickness. These lamina colors work in tandem with pigments to define the wing pattern elements that distinguish populations, sexes, and species, indicating that the ability to vary lamina thickness has been an important microevolutionary and macroevolutionary tool in this group, and likely in butterflies more broadly.

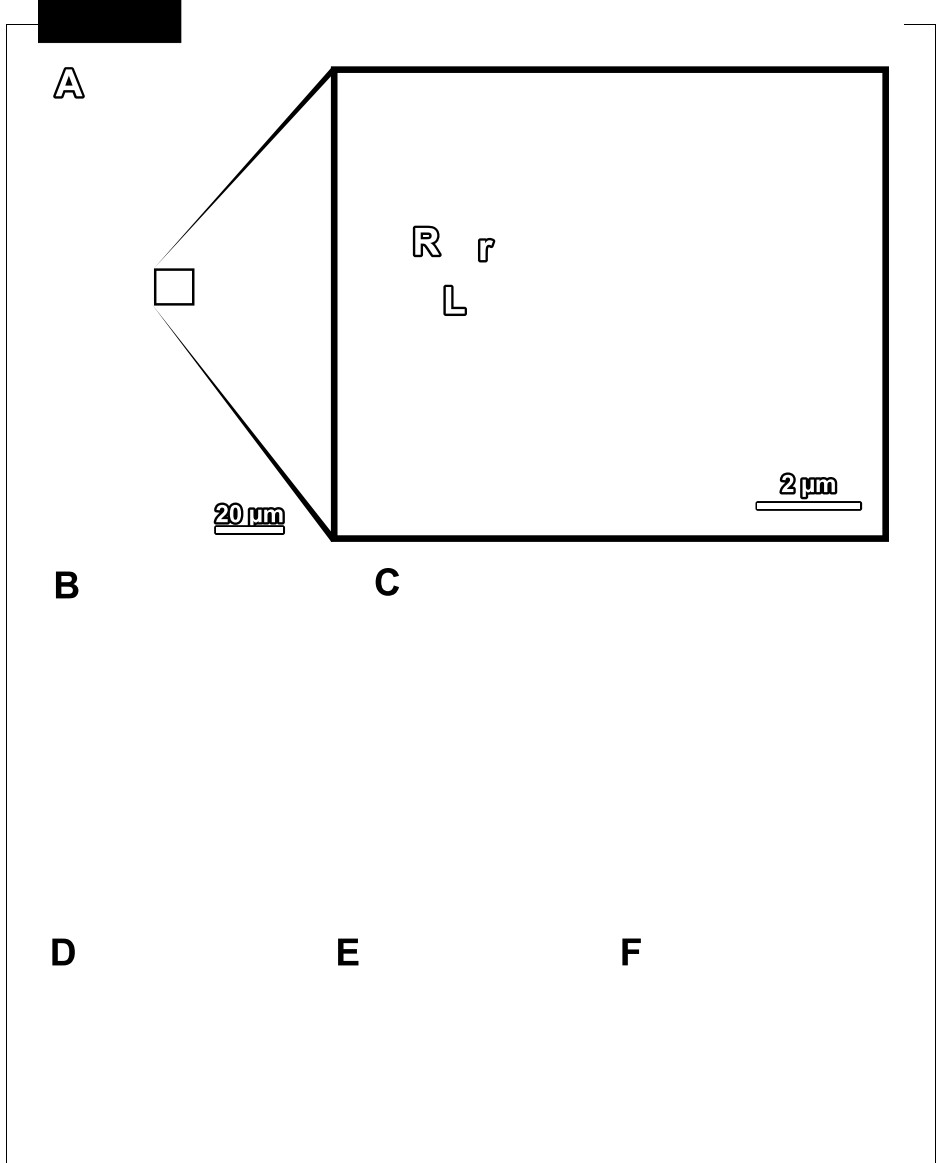

**Figure 1.** The lamina of a typical butterfly scale functions as a thin film reflector. (**A**) Colorized helium ion micrograph of a nymphaline scale, with a window milled using a gallium focused ion beam. Inset at higher magnification, with labels for general architectural components of a scale (R = ridges, r = crossribs, L = lamina). (**B**) Diagram of reflection and refraction in a chitin thin film. White light enters, reflections are produced at each surface of the film, and reflections of select wavelengths remain in phase as a function of film thickness (T). (**C**) Experimental disruptions of wing color are associated with altered lamina thickness. In *J. coenia,* artificial selection for blue color increased lamina thickness in cover scales. In an *optix* mosaic knockout mutant, certain wing regions have similar thickness increases. This trend recapitulates natural variation in *J. evarete,* where blue butterfly scales have thick laminae relative to scales from a brown individual. Boxplots show median and inner quartiles, whiskers extend to 1.5 times the interquartile range, outliers are shown as points, and notches show 95% confidence interval of the median. For wild-type and selected *J. coenia* cover and ground scales, N = 3 individuals and 20–60 measures from different lateral and distal positions within 4–11 scales. *J. evarete* and *optix* mutant data are from single individuals, with minimum N = 21 measures from three scales. (\*\* nested ANOVA p=0.00157) (**D**) Wild-type *J. coenia.* (**E**) Blue artificially selected *J. coenia.* (**F**) *J. evarete.*

Image E provided by Edith Smith

The online version of this article includes the following source data and figure supplement(s) for figure 1:

**Source data 1.** Thickness measurements for the box plot in *Figure 1C*.
**Figure supplement 1.** Wing patterning in selected blue *J. coenia.*

## Results

### Artificial selection for blue wing color increases lamina thickness

Here we describe a novel instance of rapid, artificially selected color shift from brown to blue wing color in *J. coenia* buckeye butterflies (*Figure 1D–E*) and identify the structural changes that enabled the color shift. Edith Smith, a private butterfly breeder, began selectively mating buckeyes with a few blue scales on the costal margin of the dorsal forewing (E. Smith, personal communication, Sep. 2014). After five months of selective breeding, blue spread to the dorsal hindwing of some individuals. By eight months, there was a noticeable increase in blue surface area, and within roughly 12 months (on the order of 12 generations), most butterflies in the breeding colony were visibly blue over the majority of their dorsal wing surface. On the forewing, areas proximal to M1 were visibly blue, except the discal bars (*Figure 1—figure supplement 1*). On the hindwing, blue shift did not include the distal-most wing pattern elements, namely EI-EIII and eyespots. At its strongest, the phenotype may include blue scales cupping the posterior forewing eyespot and/or a blue sheen in all distal elements of the forewing. Smith maintained the blue colony for several years, introgressing a few progeny from crosses to wild-caught buckeyes about once per year to maintain genetic diversity. Over time, she noted the emergence of a variety of short-wavelength colors, ranging from purple to green. Two years after focused selection, she estimated that the population was 85% blue, 8% green, 2% purple, and 5% brown. Like many familiar examples of human selection (e.g. domesticated animals, crop plants), outcomes are informative even without complete experimental documentation of the selective process (*Akey et al., 2010*; *Wright et al., 2005*). These selected blue buckeyes provide a previously unexploited opportunity to study structural color. They demonstrate rapid and extensive evolutionary color change, and make a stark contrast to wild-type brown populations with which they are still interfertile. Conveniently, the artificially selected taxon, *J. coenia*, is a recognized model species for butterfly developmental genetics (*Carroll et al., 1994*; *Nijhout, 1980b*). The selected blue individuals resemble naturally evolved color variants in the sister species, *J. evarete* (*Figure 1F*), and offer a useful comparison to a previously reported artificial selection experiment in butterflies (*Wasik et al., 2014*).

To pinpoint the cause of blueness in artificially selected butterfly scales, we characterized cover scales from the dorsal hindwing (*Figure 2A–D*). Butterfly wings have two classes of scales arranged in alternating rows that form two layers: superficial cover scales and underlying ground scales. Cover and ground scales frequently have contrasting size, shape, and color, and their juxtaposition can be important for wing color (*Stavenga et al., 2014*). When isolated and laid in the abwing orientation they occupy on the wing (i.e. ridges facing up), cover scales were blue (*Figure 2B*). However, when flipped over and viewed in adwing orientation, which exposes only the lower lamina, scales appeared more brightly blue and iridescence was more apparent (*Figure 2B' and D*). We tested whether the blue was structural rather than pigment-based by immersing the full scale in oil with a refractive index matched to that of chitin (*Figure 2B'''*). Index-matching eliminates the possibility of reflection and structural color, leaving only pigment-based coloration. We measured the scale's absorption spectrum under these conditions (*Figure 3A*), which revealed that blue scales did have some pigment, presumably a brown ommochrome (*Nijhout and Koch, 1991*), but this pigment cannot account for blueness. The pigment was located in the scale ridges (*Figure 2—figure supplement 1B*). Lepidopteran structural colors may occur in the lamina, lumen, ridges, or crossribs. To isolate which of these features had the nanostructure responsible for blue structural color, we dissected the scales (*Figure 2B''*, *Figure 2—figure supplement 1A*). After removing all other scale components, we found that the bare lower lamina was sufficient for blue structural color (*Figure 2B''*). We also examined regions with all scale components except the lamina and found that these pieces of lamina-less scale were not blue (*Figure 2—figure supplement 1C*). We thus focused on investigating nanostructure in the lamina. To discern between a single or multilayer lamina and take precise measurements, we cross-sectioned the lamina and viewed it with Helium Ion Microscopy (HIM) (*Figure 2C*). HIM imaging indicated the lower lamina was a simple monolayer of chitin with a thickness of 187 ± 18 nm (SD, *Figure 1C*), which is a reasonable thickness to reflect blue as a dielectric thin film (*Stavenga et al., 2014*).

We next investigated whether ground scales also contributed to blueness after artificial selection. The ground scales generally had similar architecture to the cover scales, but with less uniform lamina color: ground scales exhibited a color gradient from the stalk outward (*Figure 4A–B'*).

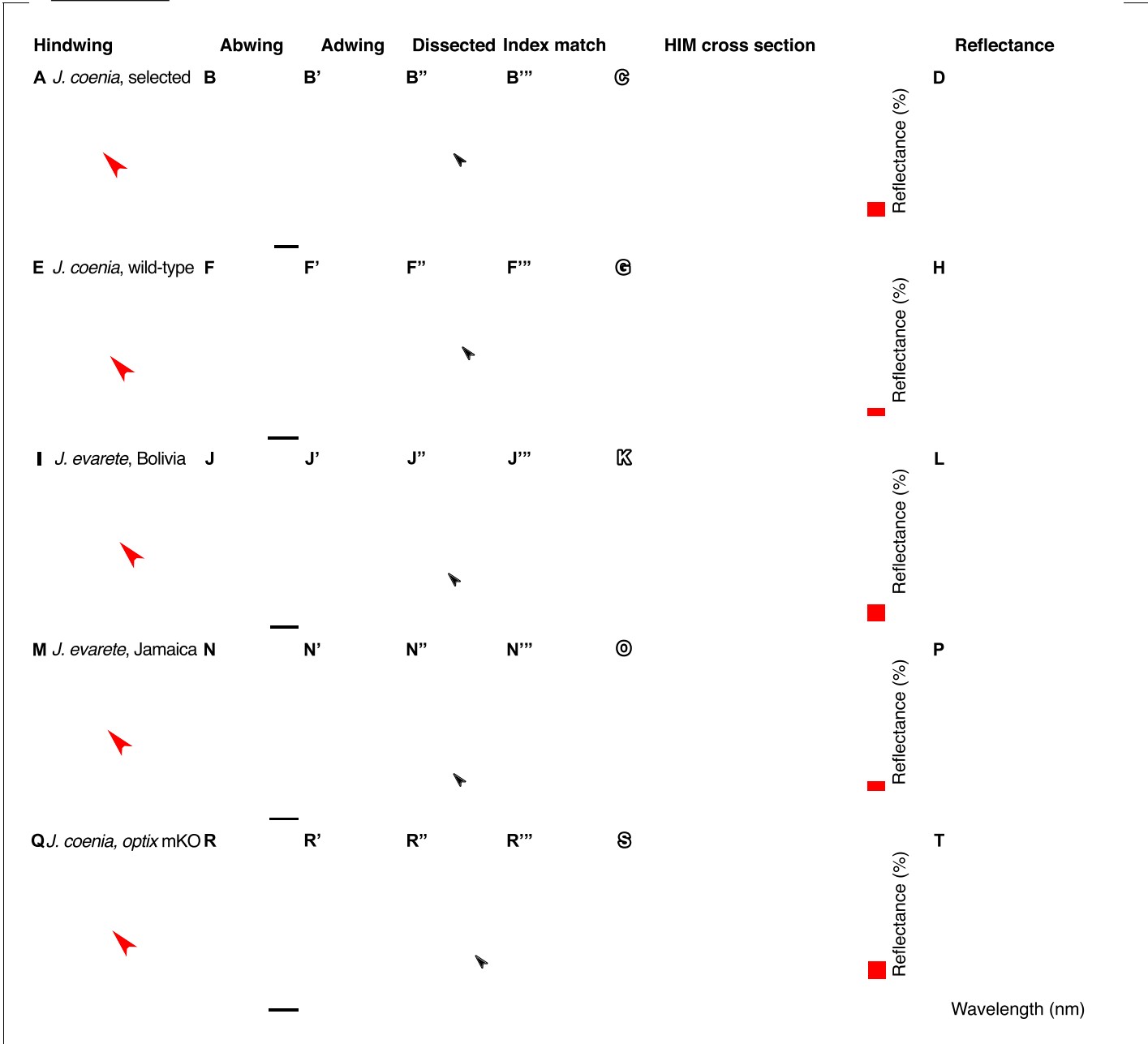

**Figure 2.** Structure and color of *Junonia* cover scales. (A–D) Artificially selected blue *J. coenia*. (E–H) Wild-type *J. coenia*. (I–L) *J. evarete*, blue male from Bolivia. (M–P) *J. evarete*, brown male from Jamaica. (Q–T) *optix* mosaic knockout mutant (mKO) in *J. coenia*. (A,E,I,M,Q) Dorsal hindwing, red arrowhead indicates the characterized scale's location. (B,F,J,N,R) Scale in the orientation it would occupy on the wing, showing the abwing surface of the cover scale. Black scale bars are 25 μm. (B',F',J',N',R') Adwing surface of cover scale, showing the underside of the lamina. (B",F",J",N",R") Dissected scale with arrow showing regions where all ridges and crossribs are removed to expose the bare lamina. The lamina is sufficient to create iridescent blue and gold structural colors. (B'",F'",J'",N'",R'") Scale immersed in fluid with a refractive index matched to chitin, thus eliminating reflection to show only pigmentary color. Blue and brown scales have comparable amounts of a brown pigment. (C,G,K,O,S) Helium ion micrograph of cross-sectioned scale. Each lamina is colorized, with approximate thickness indicated by an adjacent red bar (precise measurements were taken at sites chosen as in Materials and methods). White scale bar is 500 nm and applies to all HIM images. (D,H,L,P,T) Reflection spectra for the adwing surface of disarticulated scales. Solid line is the mean spectrum, and blue envelope is one standard deviation; minimum N = 3 scales per graph.

The online version of this article includes the following source data and figure supplement(s) for figure 2:

**Source data 1.** Processed reflectance spectra.

**Figure supplement 1.** Detailed characterization of dissected scales.

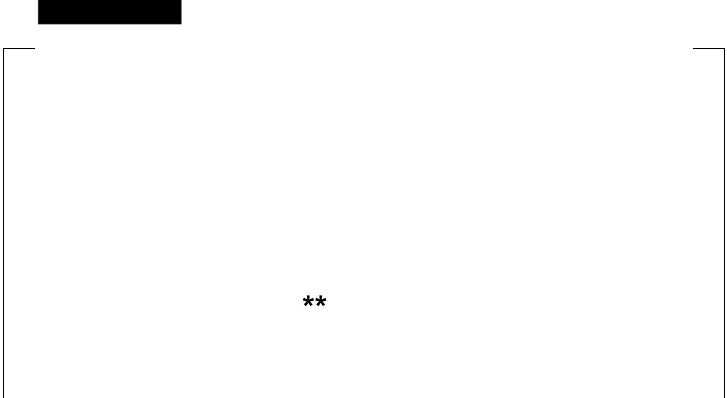

**Figure 3.** Absorbance spectra show the effect of artificial selection on scale pigmentation. (A) Absorbance measures in *J. coenia* wild-type (brown), and artificially selected (blue) cover scales show that both have comparable pigmentation (nested ANOVA, *Figure 3—source data 1*). (B) Absorbance of selected *J. coenia* ground scales is increased relative to brown wild-type scales (nested ANOVA, *Figure 3—source data 1*). (C) Absorbance of both blue and brown *J. evarete* cover scales is similar to pigmentation in *J. coenia* cover scales. Plots show mean spectra with envelope of one standard deviation, minimum N = 3 individuals and 15 scales for *J. coenia*, and N = 1 individual and six scales per *J. evarete* sample. \*\*p<0.01 at wavelengths tested in *Figure 3— source data 1*.

The online version of this article includes the following source data for figure 3:

**Source data 1.** Results of statistical tests on absorbance differences.
**Source data 2.** Processed absorbance spectra.

Correspondingly, ground scales had a similar mean thickness but more variability than cover scales (197 ± 31 nm). Ground scales were much more heavily pigmented than cover scales (*Figure 3B*, *Figure 4B"*), such that the abwing surface was black (*Figure 4B*). The extra pigmentation in ground scales enhances spectral purity by absorbing light transmitted through the cover scales, thus reducing backscatter and making the observed blue color more saturated, (similar to *Siddique et al., 2016*). We conclude that cover scale laminae are the major source of blueness in artificially selected buckeye butterfly scales, while melanic ground scales secondarily enhance spectral purity.

For comparison, we tested the source of color in wild-type brown scales and found that they also had structural color (*Figure 2E–H*). Brown cover scales had the same general architecture and no significant difference in the amount of brown pigment compared to blue cover scales (*Figure 3A*, Type III nested Analysis of Variance (nested ANOVA), *Figure 3—source data 1*). The salient difference was lamina thickness: brown scales were markedly thinner, measuring only 107 ± 14 nm (nested ANOVA, p=0.00157, *Figure 1C*, *Figure 2G*). A 107 nm chitin thin film reflects a desaturated golden color due to reflectance of many long wavelengths. This golden structural color was confirmed by the adwing scale color, the color of the bare lamina in dissected scales, and the adwing reflectance spectra of brown scales (*Figure 2F'–F", H*). Therefore, though brown coloration is often attributed to pigmentation, wild-type brown cover scales also had a structural color, one simply tuned to enhance different wavelengths.

Artificial selection also altered the ground scales relative to wild type (*Figure 4A–D*). Selected ground scales were significantly more absorbing than wild-type (brown) ground scales (*Figure 3B*, nested ANOVA, *Figure 3—source data 1*), which is consistent with increased pigmentation that decreases backscatter in blue wing regions. The wild-type ground scales were thinner on average than the blue ground scales, although the difference was not statistically significant (wild type 156 ± 33 nm; selected 197 ± 31 nm, nested ANOVA, p=0.37, *Figure 1C*). This comparison was complicated by the fact that although rainbow lamina colors in optical images suggest that each ground scale likely had highly variable thickness (*Figure 4B', D'*), we were only able to take measurements from a single cross-section position along any given scale's proximodistal axis. Consequently, our dataset seriously underestimates the within-scale variation and overestimates the between-scale variation, making this model poorly equipped to detect any differences between groups due to potential masking of the effect. A more conclusive test would require a way to thoroughly sample thickness variation within each ground scale and larger sample sizes. Our results qualitatively suggest

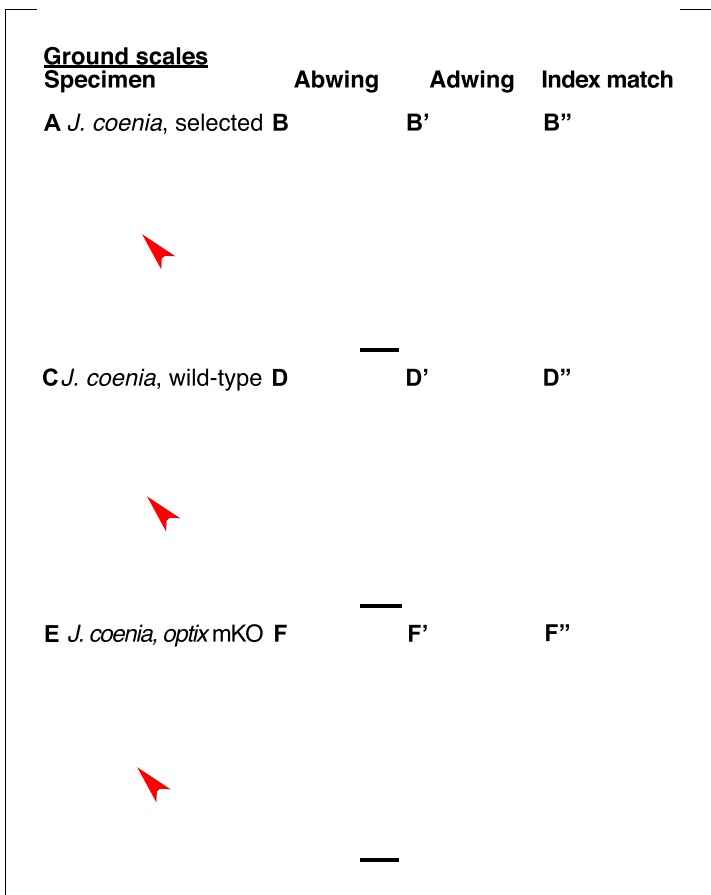

**Figure 4.** Structure and color of *J.coenia* ground scales. (A,C,E) Wings with red arrowhead indicating the region from which scales were sampled. (B,D,F) Scale in abwing orientation, that is ridges facing up. (B',D',F') Scale in adwing orientation, that is lamina facing up. (B",D",F") Scale immersed in fluid with a refractive index matched to chitin, thus eliminating reflection to show only pigmentary color. (A–B") *J. coenia* artificially selected ground scales. (C–D") *J. coenia* wild-type ground scales. (E–F") *optix* mKO mutant ground scales. Scale bars are 25 μm.

that if artificial selection increased lamina thickness in ground scales, the effect was likely less extreme than in cover scales: blue cover scales were on average 80 nm thicker than wild-type, while the observed mean thicknesses of selected and wild-type ground scale laminae differed by only 41 nm.

We conclude that the artificially selected buckeye butterflies rapidly evolved blue wing color via a 74% mean increase in lamina thickness in cover scales and, possibly, a modest increase in ground scales. The effect was further amplified by increased pigmentation in ground scales, but without removing brown pigment from cover scales. Our results show that structural color can evolve quickly by modifying one dimension of an existing structure, and the process is facilitated by the initial presence of previously unrecognized structural color in wild-type brown *J. coenia*.

Since the artificially selected *J. coenia* wing pattern resembles natural iridescent variants in the sister species, *J. evarete* (*Figure 1F*), we obtained hindwings from brown and blue *J. evarete* individuals from different geographic locations and tested whether blue cover scales in this species were also associated with increased lamina thickness (*Figure 2I–P*). We found that the same mechanism explained color differences between geographic color variants: lamina thickness distributions from brown and blue scales were non-overlapping, with blue scales having 78% thicker scale laminae on average (blue 199 ± 14 nm; brown 112 ± 13 nm; *Figure 1C*). Brown and blue scales had generally similar pigmentation, although brown scales were somewhat more absorbing at short wavelengths (*Figure 3C*, *Figure 3—source data 1*). Furthermore, in blue *J. evarete*, the ground scales were

darkly pigmented. Thus, the artificially selected blue buckeyes recapitulate natural variation at the level of scale coloration between sister species.

## Color phenotypes in optix mutants include altered lamina thickness

Recently, Zhang et al. used CRISPR/Cas9 to generate mosaic knockout mutants of *optix* (*Zhang et al., 2017*), a gene previously associated with pigment variation in butterfly wings (*Reed et al., 2011*). Surprisingly, in addition to pigmentation phenotypes, *optix* mutants in *J. coenia* gained blue iridescence in wing scales. We tested phenotypically mutant blue scales from mosaic butterflies generated by Zhang et al. to determine what structural or pigmentary changes created the color change (*Figure 2Q–T*). Where blue scales occured in the background region of the dorsal wing, blueness was due to similar factors as identified in artificially selected buckeye scales. Blue scales had markedly thicker laminae than wild-type brown scales (212 ± 11 nm, *Figure 1C*). The concentration of brown pigment in the cover scales was significantly reduced relative to wild-type scales within the same mosaic wing (*Figure 5A*, Mann-Whitney *U*, *Figure 5—source data 1*) but comparable to pigmentation in artificially selected butterfly scales (*Figure 3A*, *Figure 3—source data 1*). Ground scales (*Figure 4E–F"*) were likewise similar to selected blue ground scales, having thick and variable laminae (199 ± 31 nm, *Figure 1C*) and significantly increased pigmentation (*Figure 5B*, Mann-Whitney *U*, *Figure 5—source data 1*). Overall, blue scale identity in *optix* mutants was caused by similar mechanisms as artificially selected blue.

*o*ptix mutant phenotypes also affected structural colors and pigments differently across wing pattern elements. As originally postulated (*Zhang et al., 2017*), excess melanin was produced in some ventral wing regions (*Figure 6A–D*, *Figure 5C*). We also observed regions where both pigment and structure were dramatically changed. For example, discal bars on the dorsal forewing, which are normally orange, gained blue scales through both converting lamina structural color to blue and replacing orange with brown pigment (*Figure 6E–H*, *Figure 5D*). The kinds of pigmentation effects were diverse: *optix* mutation increased the quantity (*Figure 5B,C*), decreased the quantity (*Figure 5A*), or switched the identity (*Figure 5D*) of the pigment in different scales (Mann-Whitney *U*, *Figure 5— source data 1*). Because the butterflies were mosaic mutants, some of this phenotypic variability

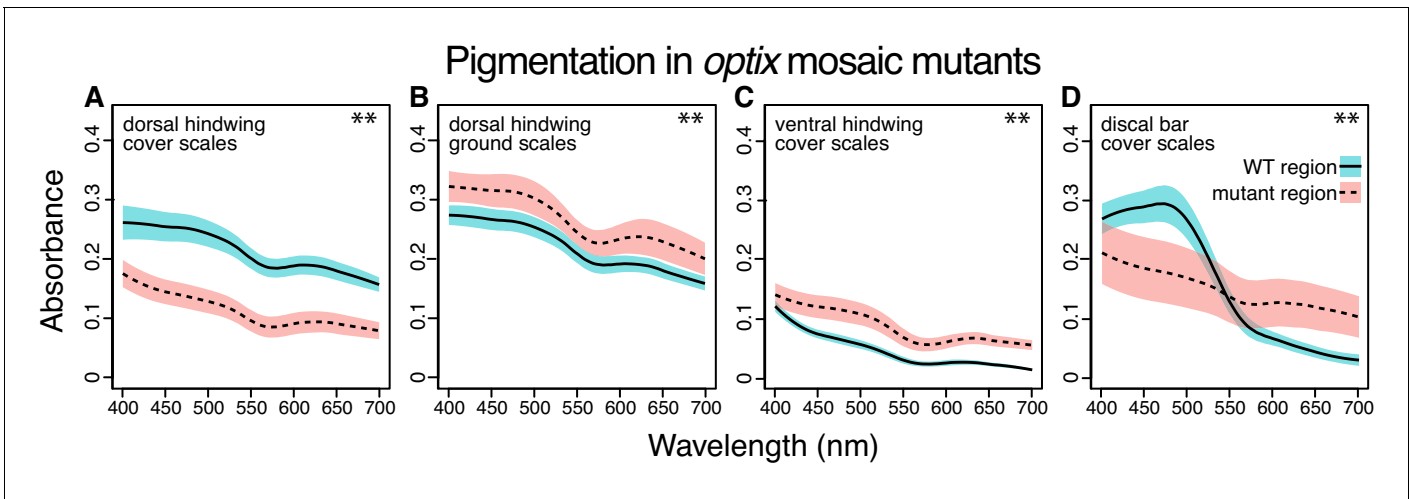

**Figure 5.** Absorbance spectra show the effect of *optix* knockout on scale pigmentation across wing pattern elements. All comparisons are between wild-type and mutant regions in the same mosaic wing. (A) *optix* mutation decreases absorption in cover scales from the main background region of the dorsal hindwing (*Figure 2Q*). (B) Absorbance of ground scales from the dorsal hindwing (*Figure 4E*) is increased in mutant scales. (C) Absorbance increases with *optix* mutation in ventral hindwing cover scales (*Figure 6A,C*). (D) In the dorsal discal bars, (*Figure 6E,G*) *optix* regulates a switch between orange and brown pigment. Plots show mean spectra with envelope of one standard deviation, N = 6 scales per sample. Differences for all comparisons are statistically significant (Mann-Whitney *U*, \*\*p<0.01 at wavelengths tested in *Figure 5—source data 1*).
The online version of this article includes the following source data for figure 5:

**Source data 1.** Results of statistical tests on absorbance differences.
**Source data 2.** Processed absorbance spectra.

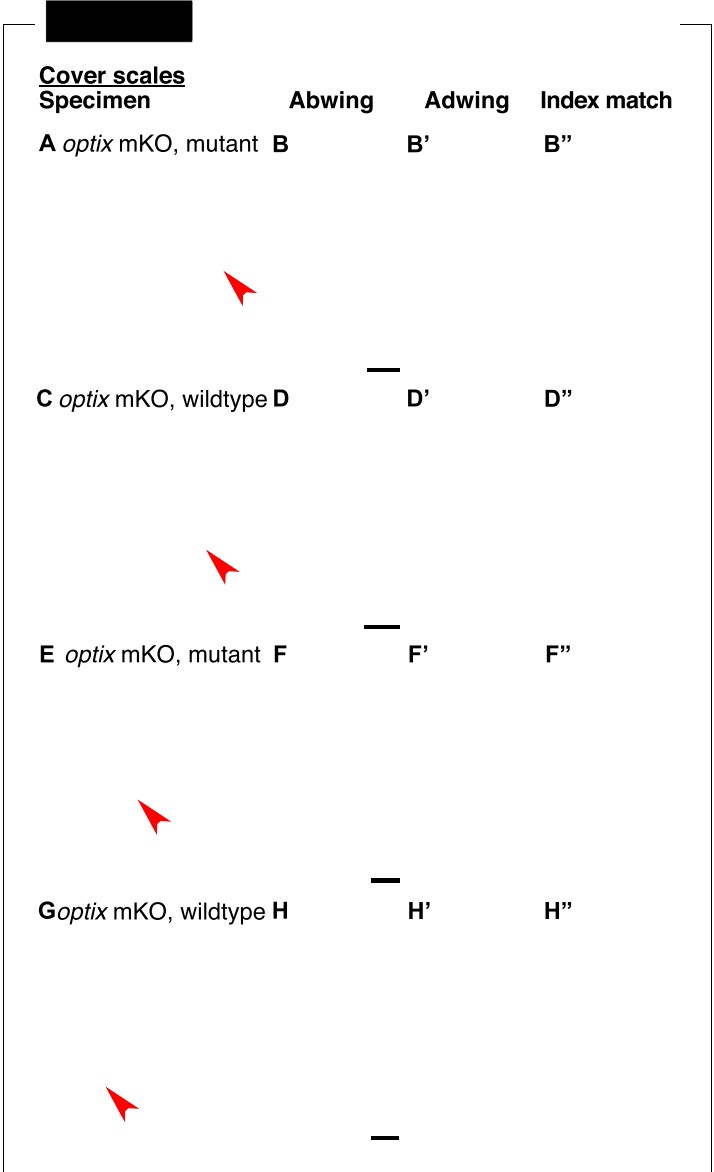

**Figure 6.** Effects of *optix* mutation on structure and color of *J.coenia* cover scales vary by wing region. (**A,C,E,G**) Wings with red arrowhead indicating the region from which scales were sampled. (**B,D,F,H**) Scale in abwing orientation. (**B',D',F',H'**) Scale in adwing orientation. (**B",D",F",H"**) Scale immersed in fluid with a refractive index matched to chitin to show only pigmentary color. (**A–B"**) Mutant cover scales from an *optix* mKO ventral hindwing have increased melanin. (**C–D"**) Wild-type cover scales from an *optix* mKO ventral hindwing. (**E–F"**) Mutant cover scales from an *optix* mKO forewing discal bar have lost orange pigment, gained brown pigment, and increased lamina thickness, resulting in a shift to blue. (**G–H"**) Wild-type cover scales from the *optix* mKO forewing discal bar have both orange pigment and an orange lamina structural color. Scale bars are 25 μm.

could be due to genotypic differences between clones (i.e. mono- versus biallelic gene deletion, as well as the exact size of the deletion) (*Zhang et al., 2017*). However, much of the variation in outcome could also be observed within single clones that spanned multiple wing pattern elements (defined by the Nymphalid ground plan [*Nijhout, 1991*], *Figure 1—figure supplement 1*), suggesting that the patterning roles of *optix* are quite context specific.

In summary, *optix* knockout can have varied effects in a single scale by altering pigmentation, nanostructures, or both. These findings are consistent with *optix*'s described role as a developmental patterning gene that determines gross switches between discrete scale fates, and which, directly or indirectly, can regulate diverse downstream factors (*Martin et al., 2014*). Since appropriate

coloration critically depends on the proper combination of pigment and structural colors in both cover and ground scales (e.g. *Wilts et al., 2011*; *Wilts et al., 2017*), it is of particular interest that *optix* can regulate all of these components simultaneously. *optix* mosaic knockout mutants demonstrate that lamina thickness can be experimentally perturbed and highlight a multifunctional candidate genetic pathway for coordinated color evolution.

## Lamina thickness consistently predicts structural color wavelength

Relatives of *J. coenia* exhibit extensive color and pattern diversity, and blue structural colors in particular show patterns of variation that hint at ecological relevance (e.g. sexual dichromatism, seasonal polyphenism) (*Figure 7A*). To assess the importance of lamina thickness variation in macroevolutionary color diversity, we sampled cover scales from nine species in the genus *Junonia* and a tenth species, *Precis octavia*, which belongs to the tribe Junoniini and exhibits seasonally polyphenic wing coloration. We prioritized large pattern elements that distinguish color forms within species. We compared scales using optical imaging, immersion index-matching, spectrophotometry, and Helium Ion Microscopy. All scales sampled had typical Nymphalid scale structure with a single plane of chitin forming the lower lamina.

We tested whether the relationship between lamina thickness and color that we observed in experimental contexts applies more broadly. We sought to address two questions: First, does lamina thickness reliably predict lamina color, as measured from the adwing surface? While it is known that the thickness of a dielectric film controls the film's reflectance, other variables such as refractive index, surface roughness, and pigmentation within the film also factor into reflectance, and these could plausibly vary among taxa. Second, how variable is lamina thickness? What range of thicknesses occur, and is there evidence for either quantized or continuous thickness variation? To address these questions, we measured reflectance spectra from the adwing surface of disarticulated cover scales from the 23 wing regions indicated in *Figure 7A*. We then cross-sectioned scales, imaged with HIM, and measured thickness.

We found that lamina thickness varied continuously between 90–260 nm, indicating that all thicknesses over a more than 2.5-fold range are accessible (*Figure 8A*). To better visualize the relationship between thickness and lamina color, we clustered similar samples into five color groups (Materials and methods). Lamina colors in these groups could be described as gold, indigo, blue, and green, with a fifth variable group that included magenta, copper, and reddish colored scales (labeled as 'red' in *Figure 8*). Thickness differed significantly between all color group pairwise comparisons (*Figure 8A*, nested ANOVA: $p < 2 \times 10^{-16}$, with *post hoc* Tukey's Honestly Significant Difference test: $p < 3 \times 10^{-8}$ for all pairwise comparisons). The color groups were also associated with different reflectance profiles (*Figure 8B*). In some cases, we obtained variable measures within individual specimens, which reflects biological color variation between adjacent scales, as well as varying color within individual scale laminae along their proximal-distal and lateral axes. A particularly striking example of the latter came from *J. atlites*. While the wing appeared light gray, at higher magnification individual scales could be seen to be multicolored (*Figure 7G'*), and thickness measures from *J. atlites* overlapped the ranges of all color groups (*Figure 8A*, see further analysis below).

Lamina thickness had a consistent relationship with adwing scale reflectance for the taxa and color range we sampled. The order of color shift as lamina thickness increased followed Newton's series, which is the characteristic color sequence for thin films (*Mason, 1927*; *Shevtsova et al., 2011*). This sequence can be understood in terms of an oscillating thin film reflectance function, which shifts toward longer wavelengths as film thickness increases (*Figure 8C–G*). The thinnest films appeared gold due to reflectance of all the longer wavelengths (*Figure 8C*). In mid-thickness laminae, a mix of two oscillations determined color: reflectance of the first oscillation was shifted toward far red wavelengths, while a second reflectance peak rose in the ultraviolet (*Figure 8D*). Visible reflectance of thicker laminae was dominated by the peak of the second oscillation as it moved from indigo to green (*Figure 8E–G*). That the trend between thickness and reflectance holds broadly suggests that color changes in *Junonia* butterfly scales have recurrently evolved via lamina thickness adjustments. Moreover, the consistency of the relationship between thickness and reflectance is useful. For example, structural variation could be rapidly surveyed by extracting fitted thickness estimates from reflectance measurements, a much less laborious process than sectioning for electron microscopy.

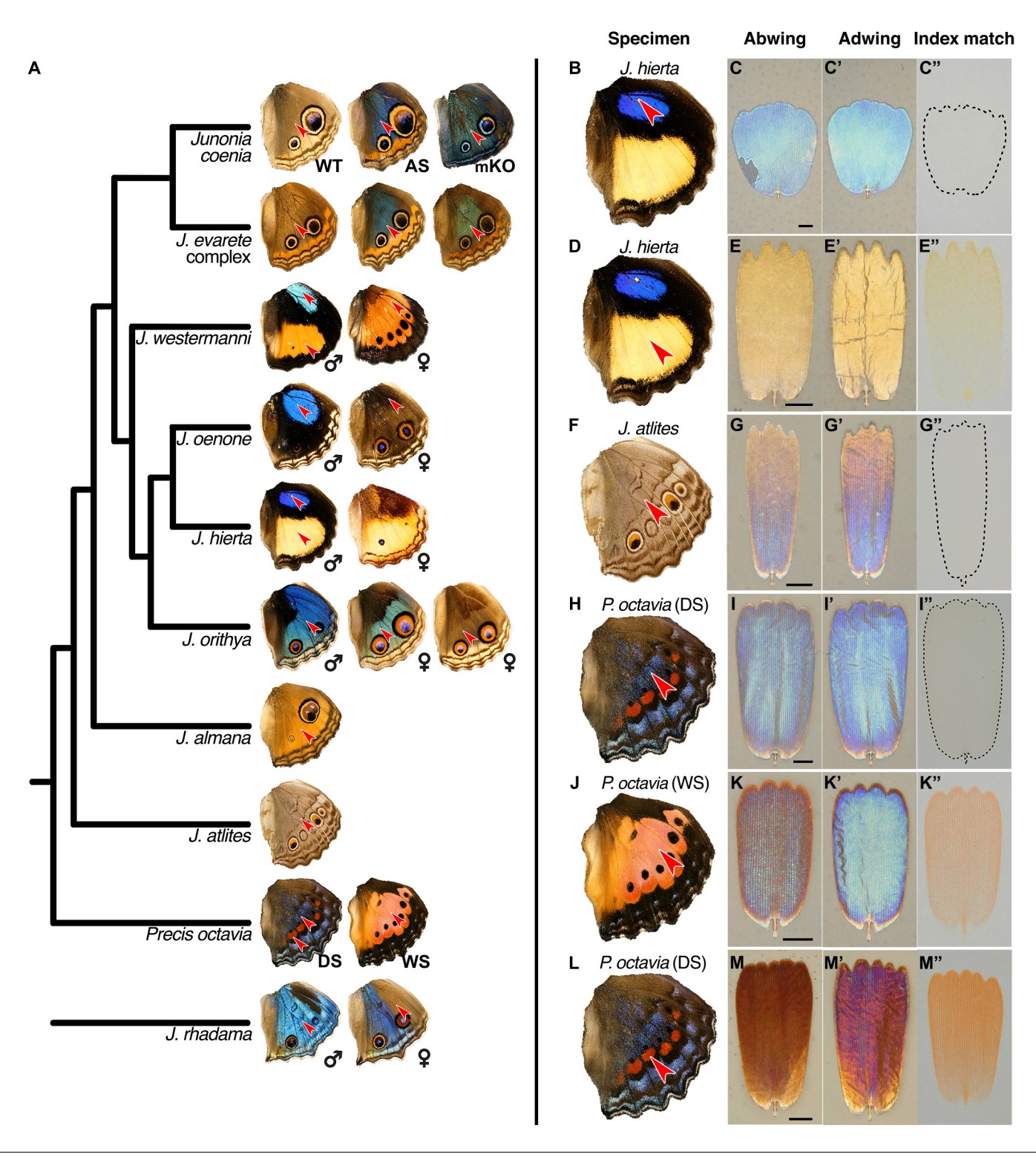

**Figure 7.** Lamina structural colors are an important component of overall wing color throughout *Junonia*. (**A**) Phylogeny of color variation in *Junonia* (based on **Kodandaramaiah, 2009**). Arrowheads indicate the color regions sampled for scale characterization. WT = wild type. AS = artificial selection. mKO = *optix* mosaic knockout mutant. DS = winter/dry season form. WS = summer/wet season form. *J. evarete* variants are from different locations. The female *J. hierta* wing image is reproduced with permission from Krushnamegh Kunte, NCBS. (**B,D,F,H,J,L**) Dorsal hindwing, arrow indicates the characterized scales' location. (**C,E,G,I,K,M**) Abwing surface of cover scale. (**C',E',G',I',K',M'**) Adwing surface of cover scale, showing lamina color. (**C",E",G",I",K",M"**) Scale immersed in fluid with refractive index matched to chitin, thus showing only pigmentary color. (**B–C"**) *J. hierta* basal aura scales

*Figure 7 continued on next page*

*Figure 7 continued*

are unpigmented and appear blue due to lamina structural color. (**D–E"**) *J. hierta* has coordinated yellow pigment with a structurally yellow lamina. (**F–G"**) Neutral light gray of *J. atlites* is exclusively structural, due to additive color mixing of the multicolored lamina. (**H–I"**) Blue scales of dry season *P. octavia* are structurally colored since no pigment is present. (**J–K"**) Wet season *P. octavia* has discordant red pigment and blue lamina colors. The red pigment is localized in the ridges and crossribs on the abwing surface of the scale, while blue light from the lower lamina spills through the windows between them. (**L–M"**) The red band in dry season *P. octavia* is a more saturated red than in (**J**), due to the combination of both more red pigment and a structurally reddish lamina.

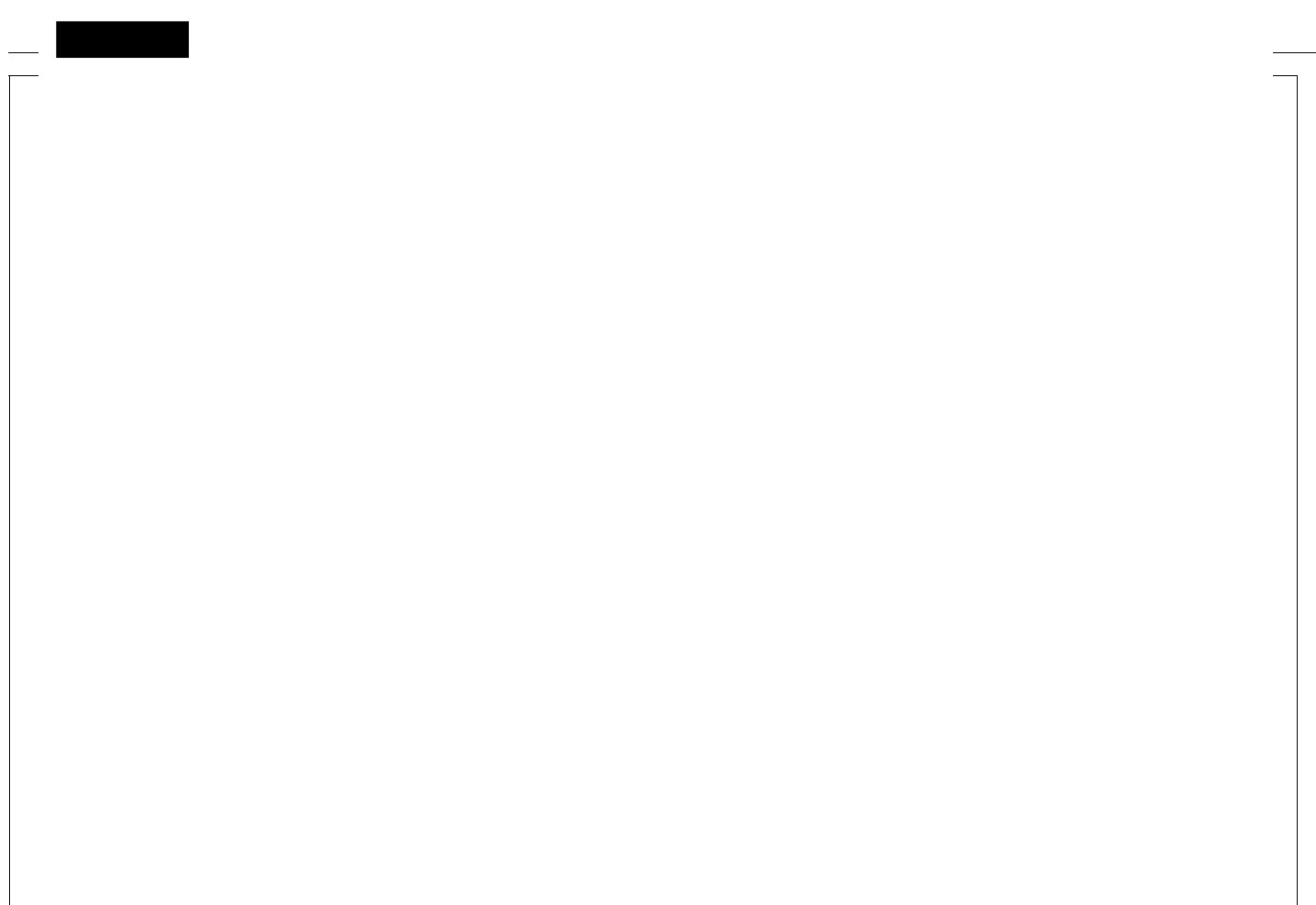

**Figure 8.** Lamina thickness predicts lamina color across the *Junonia* phylogeny. (**A**) Thickness measures for the regions indicated in *Figure 7A* vary continuously over a 170 nm range (minimum N = 3 scales and 12 measures per specimen). To visualize the relationship between thickness and color, we clustered similar specimens into five color groups described as gold, red, indigo, blue, and green. Thickness is significantly different between groups (nested ANOVA and Tukey's HSD, $p < 3 \times 10^{-8}$). *J. atlites*, which has rainbow color gradients in each individual scale, has especially variable thickness, with measures overlapping the ranges of all color groups. Boxplots show median and inner quartiles, whiskers extend to 1.5 times the interquartile range, outliers are shown as points, and notches show 95% confidence interval of the median. (**B**) Color groups are associated with different reflectance profiles. Lines are mean spectra and envelopes show one standard deviation. Minimum N = 3 scales per specimen from panel A; clusters follow panel A. (**C–G**) Adwing reflectance spectra for representative individual specimens with increasing lamina thicknesses. The color sequence follows Newton's series. Solid line is the mean spectrum and the envelope is one standard deviation; N = 3 scales and six spectra per graph.

The online version of this article includes the following source data and figure supplement(s) for figure 8:

**Source data 1.** Thickness measurements for *Figure 8A*.

**Source data 2.** Processed reflectance spectra for *Figure 8B*.

**Figure supplement 1.** Comparison of modeled and measured spectra and thickness measurement methods.

## Lamina structural color influences wing color throughout the genus *Junonia*

We next tested whether the extensive variation in lamina structural color among *Junonia* butterflies, explained by lamina thickness, also drives variation in overall wing color. An alternative hypothesis would be that composite wing color is usually dominated by pigmentation, particularly by pigments distributed on the outward-facing abwing surfaces of cover scales, above the lamina thin film. We measured pigmentation in cover scales from the same regions (*Figure 7A*) to test the relative importance of pigments and lamina structural colors for wing color. (Structural colors and pigments are listed per each specimen in *Supplementary file 1* and representative examples are shown in *Figure 7B–M"*.)

Pigmentation was highly variable among *Junonia* species (*Figure 7B–M"*, *Figure 9*, *Supplementary file 1*). This included marked differences in pigmentation between regions of a single wing (e.g. yellow and blue regions in *J. hierta*, *Figure 7B–E"* and *9A*) and also variation between color forms and species throughout the genus (e.g. between sexes in *J. orithya*, *Figure 9C*, and seasonal forms in *P. octavia Figure 7H–M"* and *9B*). Absorbance spectra varied in both shape and magnitude. Variation in magnitude, such as between the red band and the wet season morph of *P. octavia* (*Figure 9B*), represents differences in pigment abundance. We also observed distinct absorbance spectral shapes, which can indicate the identity of the pigment (for example, contrast the spectral shape of the yellow pigment in *J. hierta*, *Figure 9A*, versus red pigment in *P. octavia*, *Figure 9B*, versus brown pigment in *J. orithya*, *Figure 9C*).

Notwithstanding the clear importance of pigmentation among *Junonia* butterflies, pigment variation was insufficient to explain the breadth of wing color diversity, and lamina structural colors made up the shortfall. The importance of lamina structural color was most obvious in scales that entirely lacked pigments. For example, the blue basal aura regions of male *J. westermanni*, *J. hierta*, and *J. oenone* wings had unpigmented cover scales with structurally blue laminae (*Figure 7B–C"*, *Figure 9A*). Most of the pigmentless scales we sampled were blue, with the notable exception of *J. atlites* scales (*Figure 7F–G"*, *Figure 9D*). These scales had rainbow gradient laminae, which presumably create the overall light gray by additive color mixing (*Vukusic et al., 2009*). *J. atlites* demonstrates that lamina structural color can fundamentally drive wing color even in neutrally colored wing regions that are not obviously iridescent, and also that thickness can be patterned at fine spatial resolution within a single lamina.

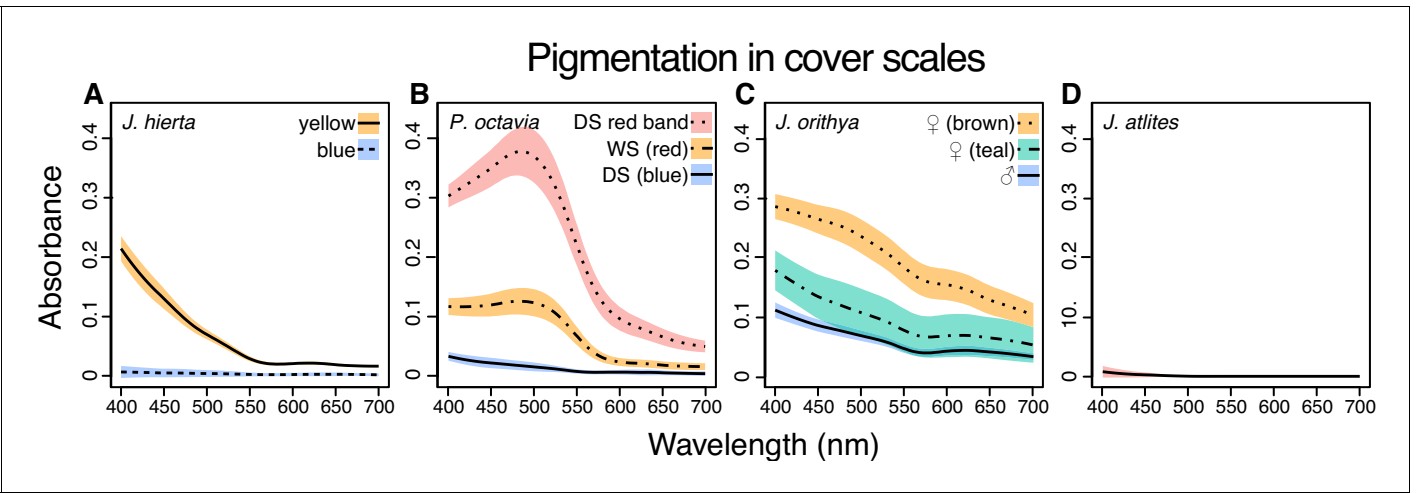

**Figure 9.** Absorbance spectra show variable pigment concentrations and identities among representative *Junonia* butterflies. Spectra were taken from cover scales from the regions shown in *Figure 7A*. (A) *J. hierta* pigmentation varies by wing region (*Figure 7B–E"*). (B) Intensity of red pigmentation is the most important driver of color difference between seasonal morphs of *P. octavia* (*Figure 7H–M"*). (C) Pigment absorbance differs by sex in *J. orithya*. (D) *J. atlites* scales lack pigmentation (*Figure 7F–G"*). Plots show mean spectra with envelope of one standard deviation, minimum N = 6 scales per sample.

The online version of this article includes the following source data for figure 9:

**Source data 1.** Processed absorbance spectra.

In most wing regions, color was determined by the interaction of both lamina structure and pigments. For example, in the cover scales of *J. hierta* (*Figure 7D–E"*, *Figure 9A*), the yellow lamina structural color and yellow pigment were mutually reinforcing, with the lamina sensibly reflecting wavelengths that the pigment does not absorb. Other examples help delineate how much pigment is required to overpower the lamina color. In blue *J. evarete*, pigments in the cover scale ridges absorbed approximately 0.2 AU (Absorbance Units, that is 37% of light not transmitted, *Figure 3C*) of the blue wavelengths that the lamina reflected most brightly (*Figure 2L*). With this ratio, wing hue was still driven by the lamina structural color. The cover scales of *J. orithya* were similar (*Figure 9C*), having a neutral dark pigment (i.e. a pigment that absorbs all visible wavelengths) in the scale ridges. Perhaps dark pigment in the ridges functions like a Venetian blind to limit iridescence, so that at high viewing angles, where iridescence would be most pronounced, light from the lamina is quenched.

Because of their range of pigment concentrations, *P. octavia* specimens were also useful to test the tradeoff between pigment abundance and lamina color influence. When viewed at high resolution, scales from the wet season morph of *P. octavia* contained red pigment in the ridges and cross-ribs (max absorbance 0.12 ± 0.02, *Figure 7K,K'*, *Figure 9B*), while reflected light from the blue lamina spilled through the windows between ridges. Viewed macroscopically, this combination made a lightly saturated red. To display a richly saturated red, much more pigment was required, as seen in the red band of the dry season morph (max absorbance 0.38 ± 0.04, *Figure 9B*, *Figure 7L–M"*). These reddest scales also had thinner, structurally magenta and copper colored laminae that may further reinforce redness (*Figure 8A*, *Figure 7M'*). The concentration of red pigment was the most important driver of the color difference between *P. octavia* seasonal morphs. The blue and red morphs had only a subtle difference in lamina thickness (*Figure 8A*), and the laminae of both were blue (*Figure 7I', K'*), but the blue morph lacked any red pigment (*Figure 9B*).

Overall, *Junonia* wing color was determined by complex mix-and-matching of different lamina thicknesses and pigments. A thin film lower lamina was present in all scales, but its influence on wing color was adjusted by the amount and placement of pigment, especially in the upper surface of the scale. Pigments can mask lamina structural color at high enough density, depending on the placement and color of the pigment as well as the color of the lamina. In our tests, when pigmentation absorbed ≤0.2 AU of the relevant wavelengths, it did not cancel out lamina structural color.

## Comparison to thin film equation

We compared our empirical data to Fresnel's classical thin film equations, which model the reflectance of an idealized dielectric thin film (*Yeh et al., 1978*; *Fresnel, 1834*). This model has previously been used to estimate the thickness of butterfly scale laminae based on their adwing reflectance spectra (*Stavenga et al., 2014*; *Wilts et al., 2017*). For each sample, we modeled the expected reflectance using our thickness measurements, and then compared to the measured reflectance spectra. We used 1.56 for the refractive index of chitin (*Vukusic et al., 1999*) and a maximal angle of illumination of 30° following *Stavenga (2014)* (because spectra were measured through an objective lens with a numerical aperture of 0.5). To account for measurement error, we modeled films over all thicknesses within one standard deviation of the measured mean per sample (red envelopes, *Figure 8—figure supplement 1A*). We also modeled films with Gaussian thickness distributions for each sample, following *Siddique et al. (2016)*. This model is analogous to a single uneven film with mean thickness and surface roughness defined by the measured thickness and sample standard deviation (solid red lines, *Figure 8—figure supplement 1A*).

We found that qualitatively the model describes the main behaviors of our data: reflectance oscillates with a given frequency and brightness, and the function shifts toward longer wavelengths as thickness increases, causing perceived color to cycle through Newton's series. Quantitatively, mean maxima and minima in the reflectance function were offset laterally for every specimen, by about 40–80 nm, with the modeled curves blue-shifted relative to the observed. A similar blue shift has been reported in butterfly scale laminae before (*Wasik et al., 2014*). The comparison improves if we assume a higher refractive index or thickness. However, to align modeled and measured spectra would require either an impossibly high refractive index (around 1.75) or increased thickness outside the error range of our measures (20–25 nm thicker than mean measurements). Possibly the lateral offset is due to a combination of the former. Alternatively, these results could indicate that scales have additional properties not fully described by the model. There are a number of differences

between the idealized film and real scales, including curvature of the film and possible birefringence of the ridges. The lamina itself may not necessarily have a uniform material composition or refractive index. For example, contrasting sublayers within the lamina (as in *Trzeciak et al., 2012*) could create extra reflective interfaces. Thus, our data are compatible with the expected behaviors of thin films, but modeling the specific case of butterfly scale laminae with quantitative precision may require additional parameters or calibration to an empirical dataset.

Modeled spectra also revealed that lamina thin films are likely UV reflective. Our spectral measurements were restricted to 400–700 nm, which roughly corresponds to the human-visible range and is adequate for demonstrating the relationship between morphology and reflectance. However, the modeled spectra show sub-400 nm reflectance of varying brightness for many lamina thicknesses that occur in *Junonia* (*Figure 8—figure supplement 1A*). Ultraviolet spectral bands are biologically relevant, since butterflies are known to have UV-sensitive photoreceptors (*Briscoe, 2008*), and this UV reflectance should be kept in mind when considering the possible ecological implications of lamina structural colors.

## Discussion

This study leverages the simplest photonic nanostructures, thin films, to interrogate the evolution and genetic regulation of structural color in *Junonia* butterfly scales. While there is a large body of literature attributing optical properties to various biological nanostructures, such claims commonly rest on correlation between mathematical models and spectral measurements. Here, we use three different experimental manipulations of the structure (dissection, artificial selection on wing color, and knockout of the *optix* gene) in addition to broad interspecies comparisons to establish that lower lamina thickness quantitatively controls structural color wavelength in *Junonia* butterfly scales. The relationship between lamina thickness and wavelength holds over a wide range of thicknesses (90–260 nm) that generate Newton's color series for dielectric thin films. Moreover, lamina structural color is one important determinant of overall wing color, including in wing regions that also contain pigments. Lamina structural colors contribute to the color differences that distinguish sexes, species, seasonal variants, and selectively-bred lineages of *Junonia* butterflies, highlighting that quantitatively tuning lamina thickness is a vehicle for color evolution in both micro and macroevolutionary contexts.

Because the lower lamina is part of the typical architecture of butterfly scales, our findings have broad implications for future research on adult color in numerous butterfly taxa. Foundational literature drew a distinction between highly derived scales with vivid structural colors and 'standard, undifferentiated scales,' which conform to the butterfly scale Bauplan, have a simple monolayer lower lamina, and 'are not truly iridescent, that is they do not produce brilliant structural colors' (*Ghiradella, 1991*). However, within the past ten years, individual examples of thin film interference from the lower lamina have emerged in diverse Lepidoptera, including in simple scales (*Stavenga et al., 2014*; *Wasik et al., 2014*; *Siddique et al., 2016*; *Wilts et al., 2017*; *Trzeciak et al., 2012*; *Stavenga et al., 2018*; *Giraldo and Stavenga, 2016*). These newer descriptions and our thorough examination of many scales indicate two points: first, although thin films are indeed less brilliant than some other classes of Lepidopteran photonic structures (thin films only reflect around 20% of incident light), they are a consequential source of structural color. Second, thin films occur in many butterfly and moth lineages and likely arose early in Lepidopteran evolution. The lower lamina has a thin film morphology in all scales that resemble the scale Bauplan, meaning that reflectance from the lamina is the shared condition except where it is masked by either heavy pigmentation or a derived structure with higher optical contrast. Because butterflies commonly produce multiple lamina colors across wing pattern elements and scale types, it is probable that the developmental genetic networks for quantitatively varying lamina thickness are deeply conserved as well. Hence, it will be useful to report which lamina colors are present, in addition to identifying pigments, when describing butterfly colors.

Physical constraints inherent to thin film colors may help explain the division of color space between pigments and photonic structures. It is not well understood why certain hues seem to be more often produced by pigments while others are more often produced by structural colors (e.g. the abundance of blue structural colors but lack of blue pigments in birds [*Stoddard and Prum, 2011*] and the rarity of one class of red structural color in birds and beetles [*Magkiriadou et al.,*

*2014*]). In *Junonia,* we show that by tuning thickness, thin film laminae can produce nearly all the spectral colors (i.e. yellow, green, blue, indigo, UV), and even light achromatic colors (e.g. light gray in *J. atlites*) via color mixing across a gradient. Yet thin films are fundamentally incapable of producing certain colors, notably dark brown, black, and pure red. The medium thickness films that most nearly approach red have inherently poor color properties due to the oscillating nature of the thin film reflectance function. Since the colors of mid-thickness films are a mix of two reflectance peaks (*Figure 8C*), they are reddish but not pure or well-saturated, and are better described as copper, magenta, and purple. Further, mid-thickness films are not bright: they reflect less total visible light than other thicknesses we observed (compare *Figure 8D–8C,E–G*). By contrast, red, black, and brown are prevalent pigment colors in *Junonia*, making pigments and thin film structural colors complementary color palettes with little overlap. The optical limitations of thin films may have partially determined how pigment families and scale architecture evolved in early butterfly lineages, which in turn initialized whether pigments or structures provide the most accessible route to evolve specific hues during subsequent diversification.

Our findings uncover a link between artificially selectable responses in lamina thickness and natural butterfly color variation, and expand on a previous artificial selection study on butterfly wing color (*Wasik et al., 2014*) which selected for violet structural color in *Bicyclus anynana*. In both *J. coenia* and *B. anynana*, color shift was accomplished by modifying the dimension of an existing structure, the lower lamina, with pigmentation being less important. Since the selected taxa diverged 78 million years ago (*Wahlberg et al., 2009*) this similarity may be informative about evolvability in nymphalid butterflies generally. However, artificial selection in *B. anynana* primarily increased thickness in the obscured layer of ground scales, which can only weakly influence color, whereas *Bicyclus* species with naturally evolved violet wing color have violet thin films in their cover scales. In our study, artificial selection continued longer (12 vs. 6 generations) and elicited a more extreme response (74% vs. 46% increase in lamina thickness). Moreover, in *J. coenia*, we show that lamina thickness increased in the cover scales and fully recapitulated the naturally evolved mechanism of structural color in the sister species *J. evarete*. The thickness increases caused a stark wing color change plainly visible by eye, with appropriate wing patterning that also resembled *J. evarete* (thickened blue scales filled the background dorsal wing, while eyespots, distal pattern elements, and the ventral wing were unaffected). Our results robustly connect a rapid microevolutionary process to macroevolutionary diversity.

By using butterflies with CRISPR/Cas9-generated knockout of the *optix* gene, we are able to provide insight into the genetic regulation of lamina thin films. It was previously known that the *optix* wing patterning gene can regulate a switch between wild-type brown and blue iridescent wing color in *J. coenia* (*Zhang et al., 2017*), but the mechanistic basis for the color switch remained unknown. Specifically, it was unclear whether *optix* regulated scale structure itself, or whether *optix* deletion merely caused the loss of brown pigment, thus unveiling a pre-existing iridescent structure. Here, we show explicitly that in certain wing regions and scale types, *optix* deletion substantially increases lamina thickness. Our findings also amend the earlier conclusion that *optix* represses structural coloration in *J. coenia* (*Zhang et al., 2017*). Rather, by regulating lamina thickness, *optix* regulates the wavelength of a photonic structure that exists in both wild types and mutants. This distinction has implications for the likely identities and behavior of downstream genetic factors, as well as the developmental basis of mutant blue coloration. For example, rather than preventing a cascade of downstream genes from acting to erect a photonic structure de novo, *optix* may subtly regulate the expression of a gene or genes that directly regulate lamina thickness, such as chitin synthase. Additionally, we uncover disparate effects of *optix* deletion on pigmentation, including promoting, suppressing, and switching the identity of pigments in different scale types. In aggregate, these results show that *optix*'s functions in *J. coenia* are highly context specific, depending on both wing region and scale type (i.e. ground or cover scale). Moreover, because *optix* can regulate both pigmentary and structural color, the *optix* pathway is an especially interesting candidate for coordinated color evolution, and further work on the detailed regulation of *optix* and its downstream targets is called for. The possibility that *optix* plays a role in patterning blue wing regions in the artificially selected *J. coenia* is especially intriguing, and motivates future investigation into whether the *optix* locus, or other loci in the *optix* pathway, were the targets of artificial selection in that population.

In summary, thin film reflectors, a morphologically simple class of photonic structures, are experimentally manipulable and broadly employed in the lower lamina of *Junonia* butterfly wing scales.

Lamina thickness explains variation in structural color wavelength, responds to selection on wing color, and is regulated by the *optix* wing patterning gene. Tuning lamina thickness facilitates both microevolutionary and macroevolutionary shifts in wing color patterning throughout the genus *Junonia*, making the buckeye butterflies a promising study system with which to decipher the genetic and developmental origins of structural color.

## Materials and methods

### Butterfly specimens

Reared *J. coenia* were fed fresh *Plantago lanceolata* or artificial diet (Southland Products, Lake Village, AK) as larvae and kept at 27–30°C on a 16/8 hour day/night cycle. Artificially selected blue *J. coenia* were purchased as larvae from Shady Oak Butterfly Farm in 2014 (Brooker, FL). Wild-type *J. coenia* were from an established laboratory colony, originally derived from females collected in Durham, North Carolina (*Nijhout, 1980a*) or were collected in California. We acquired preserved specimens from various vendors and collaborators (*Supplementary file 1*), including *optix* mutant butterflies from *Zhang et al. (2017)*. Species-level identification was generally unambiguous. However, relationships among Neotropical *Junonia* are not well-resolved and the limited molecular data available do not cleanly support current designations (*Neild and D'Abrera, 2008*; *Pfeiler et al., 2012*; *Gemmell et al., 2014*). Two recognized species, *J. evarete* and *J. genoveva*, have large ranges with extensive overlap and many variable color forms, including both brown and blue. We therefore described three Neotropical specimens as belonging to the *J. evarete* species complex to avoid accidental misidentification. Available diagnostic details, including ventral antenna club color and full collection details, are in *Supplementary file 1*.

### Optical imaging

Scales were laid on glass slides. Optical images of scales were taken with a Keyence VHX-5000 digital microscope (500-5000x lens). For refractive index matching, we used immersion oil (nD = 1.56) from Cargille Laboratories (Cedar Grove, New Jersey), and imaged with transmitted light. Scales were dissected by hand using a capillary microinjection needle. Whole wings were also imaged on the Keyence VHX-5000, using the 20-200x lens.

### Microspectrophotometry

For reflectance spectra, individual scales were laid flat on a glass slide, with the adwing surface facing up. We collected spectra of the adwing surface with an Ocean Optics Flame-S-UV-Vis-Es spectrophotometer mounted on a Zeiss AxioPhot reflected light microscope with a 20x/0.5 objective and a halogen light source. We took two technical replicates of each scale, with a minimum sample size of 3 scales per specimen. Measurements were normalized to the reflectance of a diffuse white reference ($BaSO_4$). Data were recorded with SpectraSuite 1.0 software with three scans to average and a boxcar width of 7 pixels. The software wizard determined optimal integration time from the reference sample; time was generally about 0.007 seconds. Spot size was roughly circular, 310 μm in diameter, and centered on the scale. We processed spectra in RStudio 1.0.153 with the package 'pavo,' version 0.5–4 (*Maia et al., 2013*). We first smoothed the data using the *procspec* function with *fixneg* set to zero and *span* set to 0.3. We then normalized the data using the 'minimum' option of the *procspec* function, which subtracts the minimum from each sample. Because we use a diffuse standard and scales are specular, raw spectra overestimate reflectance. We therefore followed *Stavenga (2014)* in dividing spectra by a correction factor. We used a smaller correction factor of only 2.5, because in our setup the scale does not fill the full field of view. Absorption spectra from scales submerged in index-matched oil were collected and processed similarly, but under transmitted light with an integration time of 0.01 seconds, and without the 'minimum' option.

### Helium Ion Microscopy

Surface imaging by HIM provides increased depth of field and enhanced topographic contrast compared to Scanning Electron Microscopy for a range of biological and other materials (*Joens et al., 2013*), including butterfly wing scales (*Boden et al., 2012*). Samples were prepared for HIM by laying the wing on a glass slide with the region of interest facing down, wetting with ethanol, and

freezing with liquid nitrogen. We then promptly cross-sectioned the wing through the region of interest with a new razor blade. After the sample warmed and dried, we used a capillary microinjection needle to transfer individual cut scales onto carbon tape. Scales were placed overhanging the edge of a strip of carbon tape, with one end pressed into the tape. We optically imaged the tape strip as a color reference and then transferred the tape to the vertical edge of a 90° stepped pin stub (Ted Pella #16177). While non-conductive samples can be imaged by HIM using low energy electrons for charge neutralization, we found that the unsupported overhanging edges of our scales tended to bend due to local charging (*Allen et al., 2019*). We thus sputter coated with 4.5–13 nm of Au-Pd using a Cressington 108auto or Pelco SC5. Images (secondary electron) of the sectioned scales were acquired with a Zeiss ORION NanoFab Helium Ion Microscope using a beam energy of 25 keV and beam current of 0.8–1.8 pA (10 µm aperture, spot size 4). We then used the line measurement tool in ImageJ software to measure lamina thickness from the micrographs. We corrected measurements for slight variations in working distance not accounted for by the software scale bar, using $T_{correct} = (T_{raw})/9058$ µm x $d$ µm, where $d$ is the measured working distance and 9058 µm is the reference working distance. Because these are point measures, and thickness and color vary extensively along the proximal-distal and lateral axes of individual laminae, we took measurements from multiple different positions along each cut scale. All thickness data were based on a minimum of 12 measures drawn from a minimum of 3 scales per specimen/treatment. Thickness of female *J. westermanni* scales was not measured because specimens were unavailable.

Even with vertical mounting, the sectioned surface of the scale was not always perfectly perpendicular to the direction of the imaging beam, largely due to the scales' tendency to curve. Viewing angle is critical, since measurements taken from a projected image viewed under erroneous tilt could cause systematic underestimation of thickness. We therefore tilted the microscope stage until the scale lamina was perpendicular at the measurement site, as diagnosed by observing an inflection point in lamina curvature (i.e. a switch between the upper and lower surfaces being visible). Thickness was only measured at visible inflection points (*Figure 8—figure supplement 1B–D*). We performed a tilt calibration to test the precision of our inflection point criterion and determined that an inflection point was only visible if the sample was within 4–5° of perpendicular. Since erroneous tilt is limited to 5°, thickness underestimation is limited to 1 nm. Slight overestimations are likely, due to the sputter coating. However, since sputtering was done from a direction perpendicular to the direction of measurement, sputtering primarily increased the length dimension of the scales, rather than their thicknesses. We did not adjust thickness measurements to attempt to remove slight increases due to sputtering. Note that if we had subtracted a few nanometers from each thickness measurement, then modeled and measured spectra would be offset from each other to a greater degree.

The sectioned scale shown in *Figure 1A* was milled using the gallium ion beam of the Zeiss ORION NanoFab (beam energy 30 keV, beam current 300 pA).

## Analyses

Statistical analyses were conducted in R 3.2.2. For *Figure 8A–B*, specimens were grouped following the largest natural breaks in the data for two metrics, mean thickness and weighted average reflected wavelength, which were in good agreement. For statistical tests involving lamina thickness, we used a Type III ANOVA with either treatment (*Figure 1C*) or color group (*Figure 8A*) as the fixed effect, and with the individual and scale identity corresponding to each measurement as nested random effects. We used this nested ANOVA model to account for the nonindependence among multiple thickness measurements per scale and multiple scales per individual butterfly. ANOVA was implemented with the default settings of the R package 'lme4', including its application of Satterthwaite's approximation for uneven sample sizes. For statistical tests on absorbance in *Figure 3A–B*, we used the same method except with only a single nested random effect for individual identity, since there were not multiple absorbance measurements per scale.

## Modeling film thickness

We modeled the reflectance from chitin thin films as previously described by *Stavenga (2014)*, including integrating reflectance for values of θ from zero to the maximal angle of illumination (i.e. averaging reflectances to simulate the inverted cone of light collected by the objective lens used in microspectrophotometry, given its numerical aperture). Specifically, since our objective had

NA = 0.5, we calculated reflectance over values of θ from 0 to 30˚, multiplied by 2πθ, and then averaged over the cumulative circular surface area. For the model with Gaussian thickness distributions, we followed (*Siddique et al., 2016*) using n = 400 observations from the simulated thickness distribution.

## Acknowledgements

We thank Linlin Zhang and Robert Reed for *optix* mutant wings; Karin van der Burg for *J. coenia* eggs; Masaki Iwata and Joji Otaki for *J. orithya* wings; and Krushnamegh Kunte for the image of the female *J. hierta*. We thank Edith Smith for fantastic blue buckeyes, information about their origin, and the image in *Figure 1E*. We are indebted to Ryan Null, Bodo Wilts, and Samuel Thayer for insightful discussions. Erika Anderson, Craig Miller, Michael Nachman, Chris Jiggins, and anonymous reviewers gave helpful feedback on the manuscript. Helium Ion Microscopy was performed at the Biomolecular Nanotechnology Center, a core facility of the California Institute for Quantitative Biosciences, University of California, Berkeley. Funding was provided by a National Science Foundation Doctoral Dissertation Improvement Grant DEB-1601815 (to RCT and NHP) and a National Science Foundation Graduate Research Fellowship DGE-1106400 (to RCT).

## Additional information

### Funding

| Funder | Grant reference number | Author |
|---|---|---|
| National Science Foundation | DEB-1601815 | Rachel C Thayer<br>Nipam H Patel |
| National Science Foundation | DGE-1106400 | Rachel C Thayer |

The funders had no role in study design, data collection and interpretation, or the decision to submit the work for publication.

### Author contributions

Rachel C Thayer, Conceptualization, Data curation, Software, Formal analysis, Funding acquisition, Investigation, Visualization, Methodology; Frances I Allen, Resources, Investigation, Methodology; Nipam H Patel, Conceptualization, Resources, Supervision, Funding acquisition

### Author ORCIDs

Rachel C Thayer ⓘ https://orcid.org/0000-0001-5856-7614
Frances I Allen ⓘ https://orcid.org/0000-0002-0311-8624
Nipam H Patel ⓘ https://orcid.org/0000-0003-4328-654X

### Decision letter and Author response

Decision letter https://doi.org/10.7554/eLife.52187.sa1
Author response https://doi.org/10.7554/eLife.52187.sa2

## Additional files

### Supplementary files

- Supplementary file 1. Butterfly specimens used in this study.

- Transparent reporting form

### Data availability

All data generated and analyzed in this study are included in the manuscript. Source data files were provided for Figures 1, 2, 3, 5, 8, and 9.

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
