## [Decision Letter]

**Acceptance summary:**

This paper examines the evolution of structural color in butterflies. Through various approaches such as selection, CRISPR gene knockouts, and imaging, the authors were able to find that variations in lamina thickness are the primary mechanism used to produce structural color and to tune the wavelengths that are reflected from different scale types and in different species. The authors' identification of *optix* as a gene that controls both the production of pigment and structural color is of great interest and should be further explored in future studies.

**Decision letter after peer review:**

Thank you for submitting your article "Structural color in *Junonia* butterflies evolves by tuning scale lamina thickness" for consideration by *eLife*. Your article has been reviewed by three peer reviewers, and the evaluation has been overseen by a Reviewing Editor and Diethard Tautz as the Senior Editor. The following individual involved in review of your submission has agreed to reveal their identity: Chris Jiggins (Reviewer #2).

The reviewers have discussed the reviews with one another and the Reviewing Editor has drafted this decision to help you prepare a revised submission.

The three reviewers were enthusiastic about the analysis of structural colors and the role of *optix* in this process, and in particular the role of the latter in pigmentation. They also thought that the paper was well written and clear. However, they identified several aspects of the manuscript that should be improved before publication.

Reviewer #1 would have liked to see more species tested/crossed to better show that *optix* is really the major gene involved in the process. However, after discussion with the other reviewers, it was felt that this would take too much time for a 2-month revision. I would therefore like you to simply discuss this issue in the paper.

Reviewer #2 had some difficulty with your presentation of the relationship between color and thickness, which you must clarify as to avoid any confusion.

You should also discuss the fact that structural color is likely important in the UV since most butterflies see well in these shorter wavelengths.

Finally, and although the writing of the paper is very clear, the reviewers had problems with the presentation of the figures. I suggest that you add labels to the panels and make sure that they are all present (see comment about Supplementary Figure 3).

*Reviewer #1:*

In this manuscript Thayer, Allen, and Patel examine the production and evolution of structural color in *Junonia* butterflies using directional selection, analysis of CRISPR gene knockouts, and advanced imaging methods. They find that variations in lamina thickness are the primary mechanism used to produce structural color and to tune the wavelengths that are reflected from different scale types and in different species. Interestingly they find that a single gene, *optix*, can control both the production of pigment and structural color (via modification of lamellar thickness). This work provides an in-depth look at structural color production in an easily lab-reared species using genetic tools such as directional selection and gene knockout using CRISPR, and makes comparisons across an interesting group of butterflies. This study will likely set the standard for future studies of structural color evolution and development.

The finding that *optix* is so highly context dependent is interesting. It provides a hint at underlying developmental mechanism; while it can act as a switch of relatively large effect, these results suggest that the scale-building cells still rely on combinatorial inputs from positional information to produce different scale types in the absence of *optix* expression. It would be great to know more about what those inputs are and the types of genetic or developmental changes that modify the broader patterns but this is perhaps beyond the scope of the current study.

I have one main critique: it would be fantastic to directly connect the gene that potentially controls structural color (*optix*) with the evolutionary changes observed. This might mean simply staining the wings in a species that has an expansion of blue scales to see if *optix* expression is correlated, or better yet, using crosses and genetics to determine whether the *optix* locus segregates with the presence/absence of blue scales in the artificial selection line vs. wild type. Direct experimental tests that link this particular developmental gene to the actual evolution of wing pattern would add significance and impact. It would support the link the authors draw between *optix* and wing pattern evolution by testing it directly. Any number of downstream genes might also influence lamina thickness and therefore color, and while I think it likely that *optix* is indeed responsible this should be demonstrated/tested more directly.

*Reviewer #2:*

This paper documents variation in structural colour across the genus *Junonia*. The mechanism of colour is by use of a single lamina which is therefore rather easy to characterise. The paper convincingly documents variation in lamina thickness which is correlated with structural colour. There is an impressive body of comparative data from across the genus, which is combined with CRISPR knockouts and a selection experiment. I also appreciated the emphasis on the interaction between structural and pigment colours which is not emphasised often enough. I think this comprehensive and significant body of evidence justifies publication in *eLife*. Overall the paper is clearly written.

There is a pseudo replication issue with the statistics presented in Figure 1. Individual butterflies and within an individual each individual scale is not independent. These are therefore pseudo replicated by making multiple measurements from a single scale and sampling multiple scales from an individual (subsection "Helium Ion Microscopy", first paragraph). It is not described in the Materials and methods whether there is any accounting for this in the statistics presented. If I understand correctly, each treatment is represented by a single individual in which case it is not really valid to carry out a statistical test on these data as there is no replication.

The comparison with the thin film equation is an important part of the story, supporting the hypothesis that lamina thickness is causal for colour variation. However this section is not clearly presented and I was unsure which is Supplementary Figure 3A which is the only result presented in this section (the only supplement to Figure 3 is a table). Also greater thickness is associated with a shift from brown to blue, but it is stated that 'function shifts towards longer wavelengths as thickness increases' perhaps I have misunderstood but this seems in the wrong direction.

*Reviewer #3:*

The manuscript by Thayer, Allen and Patel identifies the simplest case of structural coloration, reflection on a thin film, as an effective means of evolutionary tuning of butterfly wing coloration. Variations in scale lamina thickness lead to the creation of colorful reflections with spectral peaks and shapes that faithfully follow the underlying physical principle and are therefore relatively easy to trace, measure, model and verify. The authors convincingly show that lamina thickness is subject to artificial selection and natural evolution and may be controlled by a single gene *optix* that has also pleiotropic effects on the pigmentation in the scales. The study has been carefully designed, conducted on a very broad and sensible selection of specimens from the different taxa, is methodologically correct and leads to plausible conclusions. The graphical presentation is very good, the text has been a pleasure to read. The study itself largely builds on the existing work by Wasik et al., 2014, and Zhang et al., 2017, (both appropriately credited and cited) and as such does not present a ground-breaking novelty, but rather an incremental advance demonstrated on a very nicely chosen set of taxa, all presented in a very convincing and comprehensive way. I therefore support the publication following a couple of minor interventions.

1) I think that it is a great pity that the reflectance and absorbance data are only presented in the spectral range 400-700 nm. I understand that this is a consequence of the unlucky choice of illumination source (halogen instead of Xe arc or similar) and maybe the associated optics. However, the thinnest laminae create strong reflections in the range 300-400 nm (also shown in the model, Figure 8—figure supplement 1), which is a biologically relevant spectral band, as all studied Lepidoptera have at least one, if not multiple UV (and sometimes also violet) receptors. The absence of UV band in Figure 2H, P leads to a slightly misleading presentation which gives an impression to the reader who mentally extrapolates the spectra <400 nm that reflectance below 400 nm drops to 0, which is most likely not true. However, this is my nitpicking and does not call for additional measurements, as the authors' case is easily proven without the measurements in the UV. Anyway, I'd prefer to see my remarks implemented in the manuscript, just to remind the reader about the integral framework of the biological signaling.

2) I have been missing a technical detail in the description of the measurements of the lamina thickness: have (and how) the authors subtracted the metallic coating from the measured thickness? This may seem to be a marginal source of errors, but could also play a part in the observed systematic deviation of spectral maxima from the modeled peaks by 40-80 nm. Could the specimens also have experienced shrinking due to dehydration in the vacuum?

Please add to the Materials and methods and maybe to the Discussion.

---

## [Author Response]

Reviewer #1: I have one main critique: it would be fantastic to directly connect the gene that potentially controls structural color (optix) with the evolutionary changes observed. This might mean simply staining the wings in a species that has an expansion of blue scales to see if optix expression is correlated, or better yet, using crosses and genetics to determine whether the *optix* locus segregates with the presence/absence of blue scales in the artificial selection line vs. wild type. Direct experimental tests that link this particular developmental gene to the actual evolution of wing pattern would add significance and impact. It would support the link the authors draw between optix and wing pattern evolution by testing it directly. Any number of downstream genes might also influence lamina thickness and therefore color, and while I think it likely that *optix* is indeed responsible this should be demonstrated/tested more directly.

Yes, we agree that the possibility that the *optix* locus (or other genes downstream of *optix*) was the target of artificial selection is intriguing, and thoroughly testing this possibility is an important future direction. We have added this subject into the Discussion section, which should also further clarify that our results so far motivate future investigation into a possible role for *optix* in patterning iridescent blue in the selected blue buckeye population, but do not demonstrate such a role.

Reviewer #2: There is a pseudo replication issue with the statistics presented in Figure 1. Individual butterflies and within an individual each individual scale is not independent. These are therefore pseudo replicated by making multiple measurements from a single scale and sampling multiple scales from an individual (subsection "Helium Ion Microscopy", first paragraph). It is not described in the Materials and methods whether there is any accounting for this in the statistics presented. If I understand correctly, each treatment is represented by a single individual in which case it is not really valid to carry out a statistical test on these data as there is no replication.

Thank you for catching this oversight. We have taken the following steps to address the pseudoreplication problems with Figure 1:

First, we have taken new thickness measurements from more wild-type and artificially selected individuals and included them in the analysis. We then used a nested ANOVA which modeled individual ID and scale ID as random effects. We also switched Figure 1C from a bar graph to box plots to better convey the data. The data for *Junonia evarete* and *optix* mKO in Figure 1 still represent single individuals, owing to unavailability of specimens which can be destructively sampled and/or which have a large enough target wing area with the mutant phenotype for our cross-sectioning protocol. We have kept these data plotted in Figure 1, but have removed the statistical test. Note that the *evarete* and *optix* mKO thickness data are also included in the genus-wide test in Figure 8, where they are consistent with our general finding of a significant association between lamina thickness and lamina color.

Since the same issue—multiple measurements per individual—is also relevant to Figure 3 and Figure 8, we have made corrections there as well. For Figure 3, we similarly measured more wild-type and selected *coenia* individuals and then included individual ID as a random effect in the ANOVA model. (Unlike thickness measurements, absorbance is measured from the whole scale, so there was no nested random effect for multiple measurements per scale). For Figure 8, we re-analyzed the data using a nested ANOVA with both individual and scale ID as random effects.

We have updated the manuscript to reflect these changes in the main text as well as in figure legends, source data, the Materials and methods section, and in Supplementary file 1 of specimens used.

The comparison with the thin film equation is an important part of the story, supporting the hypothesis that lamina thickness is causal for colour variation. However this section is not clearly presented and I was unsure which is Supplementary Figure 3A which is the only result presented in this section (the only supplement to Figure 3 is a table). Also greater thickness is associated with a shift from brown to blue, but it is stated that 'function shifts towards longer wavelengths as thickness increases' perhaps I have misunderstood but this seems in the wrong direction.

The text should have directed to Figure 8—figure supplement 1A, rather than Supplementary Figure 3A. It's fixed now, thanks.

The reflectance function does shift toward longer wavelengths, but it is also a wave function, so it repeats cyclically. (One can compare with the behavior of a sine function, although the reflectance function is technically only sinusoidal when plotted against frequency.) The reflectance peak that is in the visible range in our gold samples shifts toward longer wavelengths and completely into the infrared in our thicker (e.g. blue, green) samples, while the next maximum of the function shifts from sub-visual into the blue range. Thus the function does always shift toward longer wavelengths as thickness increases, even though the human-perceived color order repeatedly cycles through Newton's color series. It is a bit confusing, since the range of thicknesses that occurred in this study span part of two cycles in Newton's series, rather than multiple full cycles. We adjusted the second paragraph of the subsection "Comparison to thin film equation" to try to explain it better.

Reviewer #3: 1) I think that it is a great pity that the reflectance and absorbance data are only presented in the spectral range 400-700 nm. I understand that this is a consequence of the unlucky choice of illumination source (halogen instead of Xe arc or similar) and maybe the associated optics. However, the thinnest laminae create strong reflections in the range 300-400 nm (also shown in the model, Figure 8—figure supplement 1), which is a biologically relevant spectral band, as all studied Lepidoptera have at least one, if not multiple UV (and sometimes also violet) receptors. The absence of UV band in Figure 2H, P leads to a slightly misleading presentation which gives an impression to the reader who mentally extrapolates the spectra <400 nm that reflectance below 400 nm drops to 0, which is most likely not true. However, this is my nitpicking and does not call for additional measurements, as the authors' case is easily proven without the measurements in the UV. Anyway, I'd prefer to see my remarks implemented in the manuscript, just to remind the reader about the integral framework of the biological signaling.

Great point, we do expect that lamina thin films produce UV reflectance as well, which may well be functionally significant for the butterflies. We have added these points into the modeling Results section.

We did try to screen for UV reflectance by imaging some of the wings with a macro camera with a UV filter. This setup worked well to detect bright UV patches on various *Colias* butterflies, but did not show UV patterns on *Junonia* wings. We do not think this rules out UV reflectance in *Junonia*, but it may not be bright enough to detect without a broad spectrum spectrophotometer and light source.

2) I have been missing a technical detail in the description of the measurements of the lamina thickness: have (and how) the authors subtracted the metallic coating from the measured thickness? This may seem to be a marginal source of errors, but could also play a part in the observed systematic deviation of spectral maxima from the modeled peaks by 40-80 nm. Could the specimens also have experienced shrinking due to dehydration in the vacuum? Please add to the Materials and methods and maybe to the Discussion.

We did not subtract any value from our thickness measurements to account for the sputter coating, for two main reasons. First, the samples were sputtered from a direction perpendicular to the direction we measured (i.e. perpendicular to the cut edge, with the scale sitting vertically in the sputter coater). Consequently, if 5 nm of Au-Pd were deposited, that primarily increased the length dimension of the scale by ~5 nm, rather than its thickness. There is presumably some small "muffin top" spillover increase to thickness from the sputtering, but we do not have a good way to estimate that increase, which would be smaller than the total depth deposited and variable between scales, depending on slight variation in the mounting angle of each scale. This possible overestimation could not account for more than a couple nanometers out of ~200 nm thickness, so it is a very small error. Second, any possible overestimation of thickness due to the sputter coating cannot explain the deviation from modeled spectra, because the error is in the wrong direction. If we subtracted a few nanometers from each thickness measurement, then modeled and measured spectra would be offset from each other to a greater degree.

Since butterfly scales are already dry, they do not require any vacuum or dehydration steps to prepare for HIM (or regular electron microscopy), so we can rule out shrinking as a source of error.

We added further details about the sputtering into the Materials and methods.